# Large-scale perspective on extreme near-surface winds in the central North Atlantic

Aleksa Stanković[1,2], Gabriele Messori[1,2,3,4], Joaquim G. Pinto[5], and Rodrigo Caballero[1,2]

[1]Department of Meteorology, Stockholm University, Stockholm, Sweden
[2]Bolin Centre for Climate Research, Stockholm University Stockholm, Sweden
[3]Department of Earth Sciences, Uppsala University, Uppsala, Sweden
[4]Swedish Centre for Impacts of Climate Extremes (climes), Uppsala University, Uppsala, Sweden
[5]Institute of Meteorology and Climate Research Troposphere Research (IMKTRO), Karlsruhe Institute of Technology (KIT), Karlsruhe, Germany

**Correspondence:** Aleksa Stanković (aleksa.stankovic@misu.su.se)

**Abstract.** This study investigates the role of large-scale atmospheric processes in the development of cyclones causing extreme surface winds over the central North Atlantic basin (30° to 60°N, 10° to 50°W), focusing on the extended winter period (October-March) from 1950 until 2020 in the ERA5 reanalysis product. Extreme surface wind events are identified as footprints of spatio-temporally contiguous 10 m wind exceedances over the local 98th percentile. Cyclones that cause the top 1% most intense wind footprints are identified. After excluding 16 (14%) of cyclones that originated as tropical cyclones, further analysis is done on the remaining 99 extratropical cyclones ('top extremes'). These are compared to a set of cyclones yielding wind footprints with exceedances marginally above the 98th percentile ('moderate extremes'). Cyclones leading to top extremes are, from their time of cyclogenesis, characterized by the presence of pre-existing downstream cyclones, a strong polar jet, and positive upper-level potential vorticity anomalies to the north. All these features are absent or much weaker in the case of moderate extremes, implying that they play a key role in the explosive development of top extremes and in the generation of spatially-extended wind footprints. There is also an indication of cyclonic Rossby wave breaking preceding the top extremes. Furthermore, analysis of the pressure tendency equation over the cyclones' evolution reveals that, although the leading contributions to surface pressure decrease vary from cyclone to cyclone, top extremes have on average a larger diabatic contribution than moderate extremes.

## 1 Introduction

The weather and climate of Europe is strongly influenced by the passage of extratropical cyclones. Cyclones are the main cause of wind and precipitation extremes during the winter season over the Euro-Atlantic sector (Fink et al., 2009; Pfahl and Wernli, 2012), and routinely generate heavy wind-related economic losses across the continent (Roberts et al., 2014). This makes extreme cyclones and the associated windstorms one of the leading natural hazards in Europe (Berz, 2005; Ulbrich et al., 2013a; Spinoni et al., 2020).

Various aspects of extreme extratropical cyclones affecting Europe have been examined in previous work, including several detailed case studies of some of the most damaging historical windstorms, like Lothar, Kyrill and Xynthia (Wernli et al., 2002; Fink et al., 2009; Rivière et al., 2010; Ludwig et al., 2014, 2015), and studies focusing on identification of common features ,from large-scale to mesoscale (see for example Earl et al., 2017, who focused on UK wind gusts by analyzing observational data), associated with extreme windstorms caused by cyclones. Hanley and Caballero (2012b) and Messori and Caballero (2015) analyzed the most salient features of the large-scale atmospheric flow in which some of the most destructive European windstorms were embedded. They showed that surface wind extremes over Europe often coincide with simultaneous cyclonic and anti-cyclonic Rossby wave breaking events in the eastern part of North Atlantic basin. Gómara et al. (2014) further demonstrated a positive correlation between Rossby wave breaking and the occurrence of explosive cyclones in the Euro-Atlantic sector. They found that the most intense cyclones in the western North Atlantic were associated with cyclonic Rossby wave breaking over western Greenland, while the most intense cyclones in the eastern North Atlantic were preceded by cyclonic Rossby wave breaking over eastern Greenland or anticyclonic Rossby wave breaking in the subtropical North Atlantic. The physical basis of these results lies in how wave breaking events influence the orientation and strength of the eddy-driven jet. Specifically, they can create favorable conditions for cyclone intensification by strengthening upper-level divergence in the right-entrance and left-exit regions of the jet core (Uccellini, 1990). Dacre and Pinto (2020) showed that interactions between Rossby wave breaking and the eddy-driven jet are also important for cyclone clustering, i.e. the passage of multiple cyclones over a fixed location within a given time period. Although the individual cyclones that pass in succession through the same region might not be extreme, the accumulated impact of wind damage and/or precipitation can be extreme when compared to individual events. Rossby wave breaking can further influence the strength and tilt of the jet, which can then steer multiple cyclones towards the same region (Pinto et al., 2014; Messori and Caballero, 2015; Priestley et al., 2017).

An approach complementary to those above takes the potential vorticity (PV) perspective (Hoskins et al., 1985). This framework evaluates cyclone evolution through the lens of interactions between positive PV anomalies at different levels and positive potential temperature anomalies at the surface, which all induce a cyclonic circulation. This perspective has been used to study individual historical storms (like Lothar in Wernli et al., 2002) and to develop climatologies of cyclones (Čampa and Wernli, 2012). By studying PV towers (i.e. positive PV anomalies vertically aligned from the tropopause to the surface) associated with extratropical cyclones in the Northern Hemisphere, Čampa and Wernli (2012) found that more intense cyclones (in terms of lower sea level pressure) are, on average, associated with more prominent lower- and upper-level PV anomalies. The PV framework was also applied to climate model simulations to study future changes in North Atlantic cyclones and near surface winds associated with them (for example in Dolores-Tesillos et al., 2022).

The hazard to life and property posed by land-falling cyclones and associated extreme winds motivates the broad literature on the topic, part of which we have outlined above. However, land-falling cyclones constitute only a small fraction of the total number of cyclones that occur over the oceanic basins. In particular, since Europe is located at the end of the Atlantic storm track, cyclone track density there is lower compared to its peak in the central Atlantic (Wernli and Schwierz, 2006; Dacre and Gray, 2009). Notwithstanding extensive research on various aspects of North Atlantic cyclones, there have been few studies specifically focused on studying cyclones that cause extreme surface winds over the ocean (de León and Bettencourt (2021)

analyzed wave heights from altimetry data, while Gentile and Gray (2023) investigated winds in the part of the Atlantic Ocean surrounding the British Isles). However, investigation of windstorms over the ocean is of interest for several reasons. Focusing on extreme windstorms over the ocean provides the opportunity to study cyclones that cause extreme 10 m winds in the region of peak cyclone track frequency. An analysis of this kind is thus useful to compare mechanisms driving extreme windstorms over the bulk of the oceanic storm track, and over Europe, which is at the end of the North Atlantic storm track. Moreover, the chosen target region provides a larger sample of intense windstorms than if focusing on land regions, which is an important aspect to consider when studying any extreme event. An additional reason for choosing an ocean region is that it removes the sometimes confounding effects of topography and land surface properties, enabling a more direct link between cyclone characteristics and surface wind footprints. On a more practical note, offshore infrastructure and busy shipping routes over the North Atlantic can be severely affected by extreme winds, resulting in sizeable insured losses (Cardone et al., 2015). Finally, strong winds drive intermittent deepening of the ocean mixed layer, affecting phytoplankton bloom dynamics in the North Atlantic (e.g. Lacour et al., 2017).

Here, we aim to address the above knowledge gap, and specifically answer the following question: What are the large-scale atmospheric factors favouring the development of extreme surface winds in the North Atlantic basin? The highest median and 98th percentiles of 10 m wind speed in the Atlantic sector occur in the central basin, approximately in the region covering 10-50° W and 40-60° N during the winter (Laurila et al., 2021b). In this study we thus focus on this region. In contrast to many earlier studies that focused on explosive cyclones or cyclone clustering in this region, we apply a bottom-up approach, whereby we first identify extreme 10 m wind events, and then study the cyclones associated with them. We investigate how these cyclones differ from weaker cyclones as regards the synoptic-scale features present during their development, their connection with the upper-level potential vorticity fields and their anomalies, as well as the strength of the eddy-driven jet with which they interact. Additionally, we perform a surface pressure tendency analysis to quantify the factors behind deepening of top-extreme and moderate-extreme cyclones. The data and methods used are described in Sections 2 and 3, respectively. In Section 4 we present results based on a composite analysis of the extreme 10 m wind events, alongside a quantitative decomposition of the mechanisms driving the deepening of the associated cyclones. The results are discussed in Section 5, where we argue that the presence of a pre-existing downstream cyclones is of critical importance for development of the extreme-wind-causing cyclones. We summarise our conclusions in Section 6.

## 2  Data

We use the ERA5 global atmospheric reanalysis from European Centre for Medium Range Weather Forecasts (Hersbach et al., 2020). We consider hourly data from 1950 to 2020 with 0.25° ($\sim$ 31 km) horizontal resolution. We analyse 10 m and 250 hPa horizontal wind components, PV from 900 hPa up to 200 hPa (18 levels) and mean sea level pressure (MSLP). It should be noted that ERA5 has known biases when it comes to 10 m wind speed; in particular, ERA5 tends to have 8-10 % lower values of the most extreme (95th and 99th percentile) 10 m winds over the North Atlantic ocean compared to satellite observations (Campos et al., 2022). However, since biases are similar across the target region of our study (Campos et al., 2022) we do not

expect these biases to affect the ranking of our events. Additionally, studies comparing extreme 10 m winds from ERA5 to observations over the continents suggest that the variability of extreme 10 m wind speeds across cyclone centres is still well reproduced even with underestimation of 10 m wind speeds, despite the much more complex topography (Chen et al., 2024). This gives further assurance that 10 m winds from ERA5 are a reliable tool to rank the most extreme wind events.

We focus only on cyclones originating in the extratropics, excluding tropical cyclones undergoing extratropical transitions (for explanation of why extratropical transitions are excluded see discussion in next section). To this end, we use post-storm analyses (best track intensity and position estimates) of Atlantic tropical cyclones from HURDAT (HURricane DATabase; Jarvinen et al. (1984)). Although HURDAT goes back to 1851, the accuracy and completeness of the dataset increase after the introduction of aircraft reconnaissance (1944 for western part of the basin) and satellites (NOAA, 2023), so its use is appropriate for the 1950–2020 period studied here.

## 3 Methods

### 3.1 Extreme 10 m wind speed event selection

We focus on the extended winter season (October-March) from 1950 until 2020. Figure 1 shows the target region, spanning $10°$ to $50°$ W and $30°$ to $60°$ N. Extreme event detection is based on a meteorological wind severity index—for brevity referred to simply as severity throughout the paper—defined following previous work on European windstorms (Klawa and Ulbrich, 2003; Pinto et al., 2012; Hanley and Caballero, 2012b). A similarly-defined index has also been applied to climate model outputs in previous work (Leckebusch et al., 2008).

Our severity index takes into consideration grid cells where the daily maximum 10 m wind speed exceeds its local 98th percentile. We calculate severity for any given day as follows. First, we find if there are any connected regions within the target domain where daily maximum 10 m wind speed has exceeded the local 98th percentile. We call such regions wind footprints. If wind footprints exist, the severity index S for each one of them is calculated as:

$$S = \sum_{i \in \text{footprint}} \left( \frac{v_i}{v_{98_i}} - 1 \right)^3 I(v_i, v_{98_i}), \tag{1}$$

where $i$ indexes all the grid cells within the connected wind footprint, $v_i$ is daily maximum wind speed at grid point $i$, $v_{98_i}$ is the local 98th percentile with respect to the extended winter climatology from 1950 to 2020, and $I(a, b) = 1$ if $a > b$ and 0 otherwise.

We calculate severity for every day in our dataset, and select days on which severity is greater than zero. We then retain the top 1% of these days for further analysis, which corresponds to 115 days. More than one wind footprint can exist for any given day over the study area. If that is the case, we take the largest severity value as the severity for that day. The reason for calculating separate values of severity for different wind footprints within the target region is that there can be multiple cyclones passing through the target region on the same day. Identifying contiguous regions reduces the possibility of attributing a footprint to the wrong cyclone. It should also be noted that if a given windstorm caused exceedances in connected regions

outside of the target region, those regions are disregarded in order to focus on the target region. A simplified illustration of our analysis procedure is shown in Figure 2.

The severity index we use was derived empirically to explain insured losses in Germany (Klawa and Ulbrich, 2003), and has chiefly been adopted for studying European windstorms. As such, it could be seen as unsuitable for the central Atlantic region. However, the index has a physical grounding since the cube of the wind speed represents the flux of kinetic energy.

The index can be used to obtain a windstorm ranking even in a context where insured losses are irrelevant. Moreover, its use of a percentile threshold makes it appropriate to study extreme winds over a region such as the central North Atlantic where climatological wind values vary markedly.

It should be noted that the severity index we use to rank event intensity is sensitive to cyclone traveling speed. In particular, it favours fast-travelling cyclones, since they have more potential to exceed local 98th percentiles in a broader area inside the

130 target region within a day. Very extreme, but slowly moving cyclones would be down-ranked. As will be shown later, top extremes are cyclones that travel rapidly because they are advected by a strong jet streak, which makes them more likely to be detected by the algorithm used here.

### 3.2 Detection and tracking of cyclones associated with the extreme 10 m winds

The basis for cyclone identification is a dataset of cyclone tracks computed using the cyclone tracking algorithm of Pinto et al.

(2005), based on Murray and Simmonds (1991), and applied to the same ERA5 data used in this study. The algorithm identifies cyclones by first finding a maximum of MSLP Laplacian (a proxy for the maximum of relative vorticity) and then finding the MSLP minima closest to it. Tracks are further filtered to exclude weak, short lived and non-developing lows by applying the criteria from Pinto et al. (2009). As was shown in Neu et al. (2013), this tracking method performs well compared to other tracking schemes and was used in numerous studies before (Gómara et al., 2014; Priestley et al., 2017, 2020; Leeding et al.,

2023).

The cyclone track dataset from Pinto et al. (2005) was refined by additionally computing tracks of cyclones associated with the top 1% of severity events using 1-hourly data, as described below. The main motivation behind this additional tracking is in the increased precision that it allows. As the tracks provided by the Pinto et al. (2005) algorithm are computed using 6-hourly data, the exact hour when the peak 10 m winds occurred could be missed by up to five hours, yielding potentially large errors

in the position of these fast-moving cyclones. We therefore refine the tracks by applying the following procedure. We first find the location of the peak 10 m wind speed within the strongest wind footprint for each day. After that, we identify the location of the cyclone associated with the event as the MSLP minimum closest to the the peak wind speed location. The identification is performed at the time of day when 10 m wind speed is strongest; in the composite analysis described below, this instant is taken as $t = 0$. We then track every extreme cyclone back in time with an hourly time-step by following the absolute MSLP

minimum. For every hour before $t = 0$, we put a box ($\pm 4°$ latitude; $+1°$, $-5°$ longitude) around the location of a cyclone at $t = t + 1$ h step in time. To remove ambiguity in cases when several MSLP minima are close to each other, we perform a Gaussian smoothing of the MSLP field with sigma of 0.1. We then look for an MSLP minimum within the box. As a check, we compare cyclone tracks obtained in this way with those produced by Pinto et al. (2005) and find no qualitative differences

(an example of the refined tracks being more precise than tracks from Pinto et al. (2005) can be seen on Figure 3b, c). We also performed a manual verification of the cyclone tracks by plotting the cyclone locations on MSLP maps (not shown). Tracks of the cyclones associated with the top 1% of 10 m wind footprints are shown in Figure 1.

The same tracking method was also used to track pre-existing downstream cyclones present after the cyclogenesis of the above top 1% cyclones. These pre-existing cyclones are tracked from the time of cyclogenesis of the extreme cyclone (yellow triangle in the example in Figure 1) up to 12 h before the peak 10 m wind speeds. Tracking after this time period proved to be less reliable since the proximity of two systems often seemed to produce multicentre cyclone-like structures (see Hanley and Caballero, 2012a).

Of the 115 events that make up the top 1%, 16 are of tropical origin and match tracks from HURDAT2, and we discard them from further analysis. The reason for discarding them is the large differences in development between purely extratropical cyclones and extratropical transitions. For example, we found that for the purely extratropical cyclones, cyclogenesis occurs around two days before the peak 10 m wind speed within the wind footprint (at $t = -2$ days), while extratropical transitions have their origin much further back in time. Additionally, fields of upper level PV and wind at 250 hPa show less coherence for extratropical transitions, thus making the composite analysis less useful. We hereafter refer to the remaining 99 purely extratropical events as *top extremes*. Top extremes are regularly interspersed through the 1950-2020 period and there is no apparent trend in the frequency of their occurrences (see Apendix). It is interesting to note that year 1999 which had many severe European windstorms (like Lothar) did not produce any events in the top-extreme class.

To assess features unique to top-extreme cyclones, we contrast them with a group of *moderate extremes*. This group consists of cyclones in the same target region but associated with the bottom 10% of events with non-zero severity. These are cyclones that cause local exceedances of 98th percentiles, but by a modest amount. To facilitate the comparison between moderate- and top-extreme cyclones, we only select those moderate extremes that had valid tracks for at least two days before the occurrence of peak 10 m wind speed in the target region (as is typical for top extremes). We find moderate extremes in tracks based on Pinto et al. (2005),as the greater number of events requires a more efficient search of cyclones identified in ERA5 compared to the tracking done backwards from the moment of maximum 10 m winds, as was done for top extremes. Since search for moderate extremes also includes a pre-defined criteria, the total number of cyclones found that satisfy it is lower than the number of moderate extreme days. At the end, we obtain 117 moderate extremes matching our criteria, a number of the same order of magnitude as the number of top extremes. Because of the difference in the detection of top extreme and moderate extreme events, we use 1-hourly and 6-hourly tracks for them, respectively.

The choice of cyclone tracking method we used could potentially impact the results. However, alternative trackings based on a different variable (like surface vorticity) should not substantially impact the tracks of top extremes. As was shown in tracking intercomparison studies (like those by Neu et al., 2013; Ulbrich et al., 2013b), different tracking algorithms tend to agree well for deeper, more developed cyclones like top extremes. A similar result was also obtained in a more recent study by Messmer and Simmonds (2021) where two different tracking methods were used to study compound extreme wind and precipitation events. Additionally, a bias towards slower-moving cyclones intrinsic to methods based on MSLP (Sinclair, 1994) should be less prominent when using 1-hourly tracks as the box within which we search for MSLP minima between the time-steps covers

a distance larger than 60-70 km which the fastest moving cyclones cover in an hour (Neu et al., 2013). Differences between the tracking methods could, however, be more important for the set of moderate extremes as cyclones in this group has a pre-defined condition of having cyclogenesis at least two days before the peak 10 m wind speeds in the target region. Since one of the biggest differences between the tracking methods lies in the identification of the time of cyclogenesis, with vorticity-based methods tending to identify cyclogenesis earlier (Neu et al., 2013), the group of moderate extremes could potentially be larger if a different cyclone tracking method was employed.

## 3.3 Composite analysis

We perform a composite analysis to study the typical large scale features associated with our two groups of cyclones. We use both cyclone- and location-centered composites to study meteorological variables of interest (like PV, MSLP and wind at 250 hPa). Because 1° of longitude spans a distance that varies with latitude, compositing on latitude-longitude regions would introduce a distortion. We thus perform the composites after regriding meteorological fields to radial grids centered on the cyclone centers, or locations of interest in the case of location-centered composites. With this aim, we apply the method from Bengtsson et al. (2007) (described in detail in their Appendix A) which has previously been used in other studies for similar purposes (for example Dacre et al., 2012; Laurila et al., 2021a; Dolores-Tesillos et al., 2022).

Most of the composite fields we show are the anomalies of meteorological fields from 1950 to 2020 climatology. To get the climatologies, we first calculate the daily means for calendar days of each variable for every grid point and at every level of interest. Then, we obtain climatologies by computing a 31-day running-mean from these data-sets.

## 3.4 Pressure tendency equation analysis

Since cyclones within the two selected groups experience surface MSLP decrease in days leading to their peak 10 m winds (as will be shown in Section 4.2), we apply the pressure tendency equation analysis to determine the main contributors to the surface MSLP decrease of top and moderate extremes between $t = -2$ and $t = 0$. This approach analyses the expanded pressure tendency equation as described in Fink et al. (2012) and Pirret et al. (2017). Most cyclones are predominantly driven by a combination of diabatic processes (radiation, latent heating) and baroclinic conversion (rising of warm air which moves polewards and sinking of cold air which moves equatorwards). The pressure tendency equation decomposes the contribution of these processes by reformulating the classical pressure tendency equation and introducing virtual temperature as the main variable.

In practice, the pressure tendency equation analysis takes six-hourly tracks of cyclones and evaluates each term of the pressure tendency equation by following a vertical column of air over a $3° \times 3°$ longitude-latitude box centered on the surface cyclone center. The equation has the following form:

$$\frac{\partial p_{sfc}}{\partial t} = \rho_{sfc}\frac{\partial \phi_{p2}}{\partial t} + \rho_{sfc}R_d \int\limits_{sfc}^{p2} \frac{\partial T_v}{\partial t} dlnp + g(E-P) + RES_{PTE},$$
(2)

where $p_{sfc}$ is surface pressure, $\rho_{sfc}$ is surface air density, $\phi_{p2}$ geopotential at the upper boundary $p_2$, $R_d$ the gas constant for dry air, $T_v$ virtual temperature, $g$ gravitational acceleration, $E$ evaporation, $P$ precipitation and $RES_{PTE}$ residuum. As the Eq.2 shows, the tendency of the surface pressure is equal to the sum of: the change in geopotential at the upper boundary (100 hPa in this study, which was found to be the most sensible choice for extratropical cyclones by Fink et al., 2012), the vertically integrated virtual temperature tendency, the mass change caused by the difference between evaporation and precipitation, and residual due to the errors from vertical integration, discretization or the data model itself. Therefore, if the column of air does not change its height, its warming will cause horizontal expansion, divergence of air and loss of mass. The end result of this process will be a surface pressure fall. Similarly, if nothing but the upper boundary of the column changes, its lowering will cause pressure decrease.

The vertically integrated virtual temperature tendency in the pressure tendency equation can be expanded in the following way:

$$\rho_{sfc}R_d \int\limits_{sfc}^{p2} \frac{\partial T_v}{\partial t} dlnp = \rho_{sfc}R_d \int\limits_{sfc}^{p2} -\boldsymbol{v} \cdot \nabla_p T_v dlnp + \rho_{sfc}R_d \int\limits_{sfc}^{p2} (\frac{R_d T_v}{c_p p} - \frac{\partial T_v}{\partial p})\omega dlnp + \rho_{sfc}R_d \int\limits_{sfc}^{p2} \frac{T_v Q}{c_p T} dlnp + RES_2, \quad (3)$$

where $\boldsymbol{v}$ and $\omega$ are the horizontal and vertical wind components, $c_p$ the specific heat capacity at constant pressure and $Q$ the diabatic heating rate. Expansion of the vertically integrated virtual temperature tendency term allows the pressure tendency equation to contain terms that represent horizontal temperature advection (interpreted as "baroclinic" contribution,the first term on the RHS of Eq.3), vertical motion (which typically cause surface pressure increases,the second term on the RHS of Eq.3) and diabatic term. Since ERA5 does not contain the diabatic heating rate products, the diabatic term (the third term on the RHS of Eq.3) is calculated as a residual from subtracting the horizontal temperature advection and vertical motion terms from the vertically integrated virtual temperature tendency, all of which are calculated explicitly. For more details about the pressure tendency equation approach, see Fink et al. (2012).

## 4 Results

### 4.1 Example of an extreme cyclone

An illustrative example from the set of top extremes is shown in Figure 3, which presents the MSLP evolution from the time of cyclogenesis until the time of peak 10 m wind speed for the selected event. Around two days before the cyclone caused the extreme 10 m winds in the target region (27 November 2018 10 UTC), the system originated along the east coast of North America as a shallow depression (Figure 3a). At the time of the cyclogenesis there was a pre-existing, well-developed cyclone situated south of Greenland. This pre-existing downstream cyclone remained in the target region during the explosive deepening of the extreme cyclone during the next two days. Once the extreme cyclone reaches the target region, it produces a large extreme wind footprint along the track (Figure 3c). During this day, the extreme and the pre-existing cyclones appear to merge, forming a broad area of low MSLP that can be classified as a multicentre cyclone (Hanley and Caballero, 2012a). The

interaction between the pre-existing and the extreme cyclone shown in this example is common to all events belonging to the
250 top extremes group (see composites in Sect. 4.2 below).

## 4.2 Composite analysis of extreme events

Composite MSLP anomalies centred on the top extremes and moderate extremes are shown in Figure 4. Like in the example above, there are pre-existing downstream cyclones to the north-east of the top extremes around the time of their cyclogenesis (Figure 4a). At this time both lows have anomalies of up to 10 hPa. As time proceeds, the top extreme cyclones deepen and
255 approach the pre-existing downstream cyclones. At the time of peak 10 m wind speeds ($t = 0$, Figure 4c), the two systems have merged in the composite. Negative MSLP anomalies also reach their minimum, with anomalies exceeding -35 hPa. The rapid deepening of top extreme cyclones that occurs as they approach the pre-existing downstream cyclones is in line with the result that majority of top extremes (87 out of 99 - or 88%) are explosively deepening cyclones with the normalised values of 24-hourly pressure decrease greater than 24 hPa (the criteria used in Sanders and Gyakum, 1980).

On the other hand, composites of moderate-extreme cyclones (Figure 4d-f) reveal an absence of pre-existing downstream cyclones at $t = -2$ days. In fact, a weak positive MSLP anomaly with values lower than 5 hPa is found in the region where pre-existing cyclones are present for top extremes. Therefore, the presence of the pre-existing downstream cyclones appears to be an essential feature in generating top extremes.

Top extreme cyclones are also associated with an anomalously strong jet streak from $t = -2$ to $t = 0$ days (Figure 4a–c). The
265 cyclones cross the jet streak from $t = -2$ days, when they are located around the right-entrance region of the jet, to $t = 0$ days when they are in the left-exit region of the jet. Right-entrance and left-exit regions of the jet streak are associated with strong upper level divergence, which makes them favourable for the intensification of cyclones (see for example Uccellini, 1990; Rivière et al., 2010). The absolute values of composite wind speed at 250 hPa for top-extremes are over the broad regions where the wind speed exceeds 40 ms$^{-1}$ (Figure 4a–c).

Figure 5 shows the evolution of upper-level fields corresponding to the surface composites in Figure 4. Positive upper level PV anomalies associated with top-extreme and pre-existing downstream cyclones after the cyclogenesis are shown in Figure 5a–c. Two days before the peak 10 m wind, there is a well-defined, zonally-extended area of positive PV anomalies stretching to the north-east of the developing extreme cyclone (Figure 5a). At the same time, wind speed anomalies at 250 hPa show cyclonic upper level winds around the pre-existing downstream cyclone. This cyclonic flow is organised so as to
advect high-PV air southward, helping promote positive PV anomalies at the location of the top extreme cyclone. As the top extreme cyclone moves closer to the pre-existing downstream cyclone, positive upper-level PV anomalies associated with the two systems merge into a broader area of statistically significant positive PV anomalies. The intensity of the anomalies increases from $t = -2$ to $t = 0$. At $t = 0$, when the two composite systems have fully merged and surface winds reach their peak, upper level PV anomalies reach a maximum of over 3 pvu.

The moderate extremes (Figure 5d–f) display a small downstream region with positive upper-level PV anomalies, yet these are not statistically significant, and are not connected with a surface low. Large positive upper-level PV anomalies are confined to the area around the extreme cyclone itself, and never exceed 2 pvu even at $t = 0$. Consistently, wind speed anomalies at

250 hPa (Figure 5d–f) are weaker than those shown in Figure 5a–c, and substantial anomalies are only present in the region around the surface low. Absolute values of the jet streak at 250 hPa are much weaker for moderate- than for top-extreme cyclones (Figure 4), making the jet streak less able to facilitate intensification of the cyclone. It should also be mentioned that PV anomalies at $t = 0$ days averaged in a circle with a radius of 300 km around the cyclone centres reveal that two groups also differ in the strength of the lower-level PV anomaly. Top extreme cyclones have a larger median of positive PV anomalies from 900 hPa to 200 hPa (see Appendix) which agrees with the previous findings by Čampa and Wernli (2012).

To investigate the conditions leading to the development of extremes prior to their time of cyclogenesis, we compute lagged composites centered on the location of cyclogenesis (Figure 6). At $t = -6$ days before cyclogenesis, the composite for top extremes (Figure 6a) shows a broad lobe of high PV directly to the North of the cyclogenesis location. Considering that cyclogenesis in all cases occurs near the east coast of North America, this lobe is consistent with the regional PV climatology, which features a high-PV lobe over the Hudson Bay (Figure 1; mean location of top-extreme cyclogenesis is marked by a green cross). The composite PV field also displays a moderate positive PV anomaly to the east of the cyclogenesis location, which corresponds to a surface MSLP anomaly. Both persist and strengthen in subsequent days (Figure 6b,c), and can be identified as the pre-existing cyclone discussed above. This upper-level PV anomaly results from a deformation of the climatological PV structure reminiscent of cyclonic Rossby wave breaking, which has been robustly identified as a precursor of extreme North Atlantic cyclones in previous work (Hanley and Caballero, 2012b; Gómara et al., 2014).

In addition, at $t = -6$ days a positive PV anomaly appears to the west of the cyclogenesis location. This anomaly has no surface footprint, suggestive of an open-wave upper-level anomaly which in subsequent days propagates eastward until it reaches the location of cyclogenesis off the east coast of North America. Thereafter, it becomes the upper-level component of the extreme cyclone (Figure 5a–c). In fact, from $t = -3$ days before cyclogenesis (Figure 6b), a band of positive PV anomalies stretches between the incipient top extreme cyclone and the pre-existing downstream cyclone. This band can be identified after cyclogenesis as corresponding to the jet streak seen in Figure 4a.

Turning to moderate extremes (Figure 6d–f), we see that six and three days before cyclogenesis there are no statistically significant MSLP anomalies, while upper-level PV anomalies are confined to small regions and anomalies of wind speed at 250 hPa are much weaker and less organised compared to top extremes. MSLP and upper level PV and wind anomalies only strengthen one day before cyclogenesis (Figure 6f) and are associated with the moderate extreme cyclone itself: while there is a region of positive PV anomalies to the east of the incipient moderate extreme cyclone, it has no surface footprint and is absent in subsequent days. Thus, as discussed above, the main difference between top and moderate extremes is that the latter lack a well-organized, persistent pre-existing downstream cyclone to the east of the incipient cyclone.

### 4.3 Pressure tendency equation analysis

In this section we take a different perspective and quantitatively compare the mechanisms leading to deepening of top and moderate extremes using the surface pressure tendency decomposition approach. Figure 7 shows contributions of each term in the pressure tendency equation averaged over the two days leading to the peak 10 m wind speed for both cyclone groups. The vertical velocity term leads to surface pressure increase, i.e. to a weakening of the surface cyclone. Horizontal temperature

advection—the baroclinic term— is negative (strengthens the cyclone) and slightly smaller in magnitude than the vertical velocity term. The absolute values of both terms are larger for top extremes than for moderate extremes (Figure 7), which is one difference between the groups.

The largest difference between the groups lies in the pressure decrease caused by the diabatic term. The diabatic term for moderate extremes is around half of the baroclinic term. For top extremes, the diabatic term is as large as the baroclinic term. With both terms being larger than for the moderate extremes, the total surface pressure drop for top extremes is around twice as large. For all of these terms (vertical, baroclinic, diabatic and total pressure decrease) the mean values of top extremes are statstically different from moderate extremes at the 95% confidence level according to the Wilcoxon signed rank test.

Figure 7 also shows that other terms in the pressure tendency equation (the geopotential term and the term that arises from changes in mass due to the difference between evaporation and precipitation) are on average of minor importance for cyclone intensification. The residual for the selected storms is also close to zero, implying that the decomposition accurately captures the drivers of the observed surface pressure drop.

On an individual storm level, the pressure tendency equation analysis shows a large variability in the influence of the different
terms, as can be seen from uncertainty ranges in Figure 7. Even considering uncertainties, common features for both groups are a small influence of evaporation/precipitation term, a vertical term which increases the surface pressure and the dominance of either baroclinic or diabatic terms for cyclone development. Which one of the latter two is the leading driver of surface pressure fall, however, changes from storm to storm. This is in line e.g. with the results of Pirret et al. (2017) who found a wide range for the different contributions for cyclone deepening.

For top-extreme cyclones, a slightly larger group of cyclones is primarily driven by diabatic processes compared to baroclinic processes (41 versus 35, respectively). The rest of the cyclones in the group have a difference between diabatic and horizontal temperature advection smaller than 5% (the criterion we used to identify predominantly diabatic or baroclinic storms). Separate composites for diabatic versus baroclinic cyclones show no evident qualitative differences from the composites for all top extremes, even when imposing a 10% difference between the two terms to identify the cyclones (not shown).

As the top extremes occur over a period under which the signal of climate change has become more prominent, it is relevant to investigate potential trends in the surface pressure tendency terms. To do this, we divide the events into two sub-periods—one from 1950-1985 and other from 1986-2020—with the difference between the two periods providing evidence for a warming-related trend. Comparison of pressure tendency equation analysis for two periods shows that there is a statistically significant increase of the diabatic contribution to surface pressure decrease and of the total surface pressure decrease in the warmer
period (at the 95% confidence level according to the Wilcoxon signed rank test) (see appendices). There is also an increase in the baroclinic contribution, though it is not statistically significant. Moderate extremes, on the other hand, do not show any significant changes between the periods, suggesting that the most extreme cyclones are most sensitive to warming.

## 5 Discussion

The above analysis shows that the presence of a pre-existing downstream cyclone is the key feature distinguishing top-extreme from moderate-extreme cyclones. A composite analysis of top-extreme cases shows a gradual build-up of positive upper-level PV anomalies to the North of the cyclogenesis locations. After cyclogenesis and before the peak 10 m winds, top-extreme cyclones cross the jet streak while approaching the pre-existing downstream cyclone. At the time of peak 10 m winds, there is a merging of top-extreme and pre-existing cyclones, as their MSLP and positive upper-level PV anomalies form a joint large-scale system. On the other hand, the development of moderate-extreme cyclones generally occurs in the absence of pre-existing downstream cyclones, both before and after their cyclogenesis. The jet and the upper-level PV anomalies are weaker for moderate-extreme cyclones, as are the negative MSLP anomalies at the time of their peak 10 m wind speeds.

Pre-existing downstream cyclones may favour the intensification of top-extreme cyclones in at least two ways. One is through the intensification of the jet streak. Upper-level PV composites show a pattern reminiscent of cyclonic Rossby wave breaking in the days before the genesis of top-extreme cyclones (Figure 6b). Before cyclogenesis, pre-existing downstream cyclones are situated just to the east of the climatological high-PV reservoir centered over the Hudson Bay. Wind anomalies at 250 hPa associated with pre-existing downstream cyclones favour southward advection of high PV air, generating positive PV anomalies, strengthening PV gradients and generating strong jet anomalies. As shown in previous work, positive jet anomalies are associated with rapidly intensifying cyclones over the North Atlantic (e.g. Gómara et al., 2014) due to strong upper-level divergence in the jet's right-entrance and left-exit regions. Another reason why the presence of pre-existing downstream cyclones could be important is through direct cyclone-cyclone interactions and merging of the extreme and pre-existing cyclones. In particular, Figure 5a–c suggests that the two cyclones become intertwined and combine their PV to yield a very strong upper-level PV anomaly, which is typically associated with very intense MSLP anomalies (Čampa and Wernli, 2012). The precise dynamics of this merging process, and of the cyclone-jet streak interaction mentioned above, provide interesting avenues for future research.

The situation where one cyclone develops to the south-west of another pre-existing downstream cyclone, as occurs for the top extremes, is reminiscent of cyclone families and secondary cyclogenesis, a concept originating from Bjerknes and Solberg (1922). Composite analysis, however, is not the best tool to judge whether top-extreme cyclones are secondary cyclones because of the smoothing of fields intrinsic to the compositing. Answering this question would require investigating each extreme cyclone individually, and assessing its relation to a trailing cold front generated by the pre-existing cyclone (for example using the metrics of Priestley et al., 2020), which is another possible direction for future work.

We also analysed the development of top- and moderate-extreme cyclones using the pressure tendency decomposition framework. Despite large storm-to-storm variability in the relative size of the terms in the pressure tendency equation, in line with Fink et al. (2012) and Pirret et al. (2017), comparison of the factors that drive the deepening of storms between the two cyclone groups identified here reveals a systematically greater influence of the diabatic term for top-extreme cyclones than for moderate cyclones. All the leading terms have greater absolute values for top-extreme cyclones, but the diabatic term has a greater relative importance as well compared to the baroclinic term. One possible explanation could be that stronger storms have greater vertical velocities, which all else equal would imply an increase in condensation rates and in diabatic heating.

However, it remains unclear whether the increase in the absolute baroclinic contribution (that is also seen for top-extreme cyclones) is driving the increase in the diabatic contribution, or if the opposite is the case.

There are several ways in which this study could be expanded to further understand cyclones that cause extreme 10 m wind over the ocean. Investigating whether the mechanisms identified here are important in other ocean basins (North Pacific and Southern Ocean) is an obvious next step. This could also include investigation of long-term trends in storminess, along the lines of Feser et al. (2015). To quantitatify connections between the PV anomalies, 10 m wind speeds and cyclone-cyclone interactions identified here, a different kind of analysis would need to be performed, as PV inversion analysis or idealised modeling studies. Additionally, near surface winds occur in the boundary layer, which makes them a multi-scale phenomenon involving mesoscale and turbulence-scale processes. As this study only investigates large-scale dynamics that favour extreme 10 m winds, one route of further research could delve deeper into the mesoscale processes associated with these systems to provide linkages that connect large-scale with boundary layer physics. Such studies could implement tools used to detect sting-jets (like, for example, those in Manning et al., 2022; Hart et al., 2017), a mesoscale phenomenon that has repeatedly been linked to extreme near-surface winds (see for example Hewson and Neu, 2015) and not addressed in this study. Future studies with a focus on boundary layer processes could also investigate how mechanisms known to influence PV in the boundary layer, like creation of PV with latent heating along the warm conveyor belt or destruction of it through heat fluxes in the cold sectors (as shown in Plant and Belcher, 2007) act in the case of top-extreme cyclones.

## 6 Conclusions

We provide a large-scale perspective on extreme near-surface winds in the central North Atlantic. We select cyclones associated with the top 1% of extreme 10 m wind events during boreal winter, top extremes, and compare them with a group of moderate extremes—cyclones that also cause strong winds but with weaker footprints. We analyse both groups of cyclones through time-lagged composites and through the surface pressure tendency decomposition. We aim to determine the large-scale circulation features favouring the development of top extremes. We find that a key feature of top extremes is the presence of a pre-existing cyclone to the northeast of the developing cyclone. These pre-existing downstream cyclones can be identified at least 6 days prior to genesis of top extreme cyclones, but are generally absent for more moderate extremes.

The genesis of top extreme cyclones occurs around two days before they reach peak severity. The pre-existing downstream cyclones help to generate a jet streak to the east of the incipient top extreme cyclones. As top extreme cyclones develop, they cross this jet streak and experience explosive deepening and intensification of upper level PV anomalies. The pressure tendency equation analysis shows that the main difference between these top extremes relative to moderate extremes ones is the significantly larger median contribution of diabatic processes to cyclone growth in top extremes. Although there is a large variation in the relative role of the terms contributing to surface deepening from storm to storm, all the leading terms in the pressure tendency equation have, on average, larger absolute values for top extremes.

*Code and data availability.* The ERA5 reanalysis data used in this study can be downloaded from https://cds.climate.copernicus.eu/cdsapp# !/dataset/reanalysis-era5-pressure-levels?tab=form (last access: 26 March 2024) (Hersbach et al., 2020). We have also downloaded parts

of ERA5 data from Research Data Archive at the National Center for Atmospheric Research, Computational and Information Systems Laboratory. https://doi.org/10.5065/BH6N-5N20 (last access: 29 April 2024). Storm tracks found in ERA5 by using the algorithm from Pinto et al. (2005), as well as the code used for this analysis, are available upon request.

*Author contributions.* AS, RC and GM designed the study. AS carried out the analysis and drafted the first version of the manuscript. RC, GM and JGP provided methods and data. All authors contributed with discussions, structuring the analysis and reviewing the manuscript.

*Competing interests.* The authors have no competing interests to declare.

*Financial support.* This research has been supported by the Horizon 2020 framework programme H2020 Excellent Science (Marie Skłodowska-Curie grant agreement no. 956396, EDIPI project). JGP thanks the AXA Research Fund for support.

*Acknowledgements.* We thank the two anonymous reviewers for their constructive comments on the manuscript.

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

540

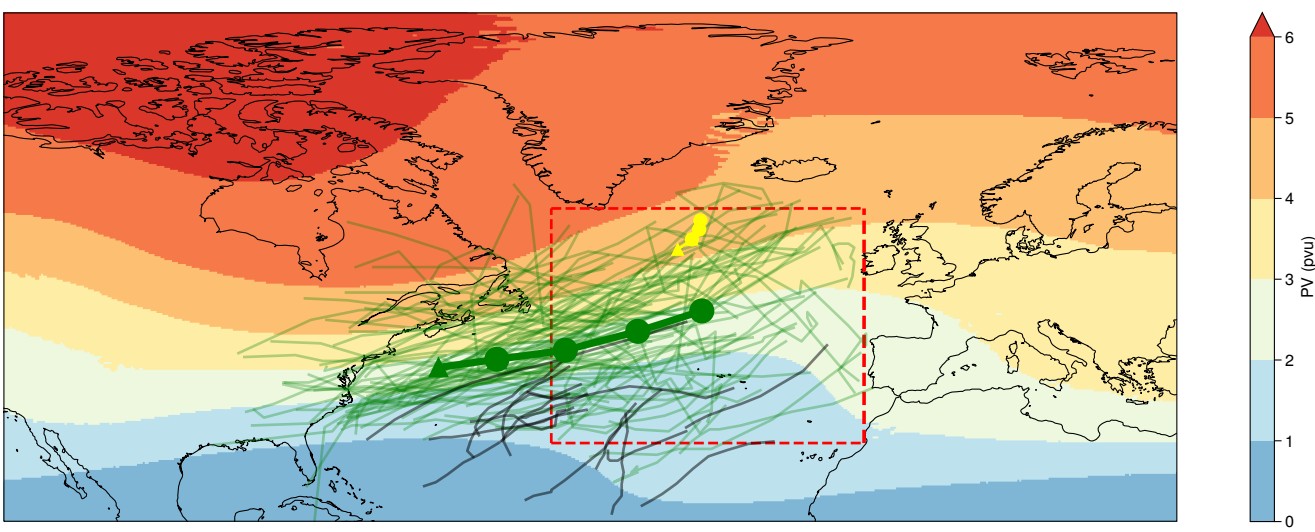

**Figure 1.** Upper-level PV (200-300 hPa mean) climatology for extended winter season (Oct-Mar) from 1950 to 2020 (colors). The red box shows the target region used to study windstorms. Storm tracks of extratropical/top-extreme cyclones (green lines) and tropical cyclones (black lines) associated with the top 1 % 10 m wind severity events from two days before until the time of maximum 10 m wind speed. Average 12-hourly track of top extremes is shown as a green thick line, with the green triangle representing the mean location of cyclogenesis of top extremes. The average storm track of the pre-existing downstream cyclones from two days to 12 hours before the peak 10 m winds is shown as a yellow line, with the yellow triangle showing their mean location at the time of cyclogenesis of top-extremes.

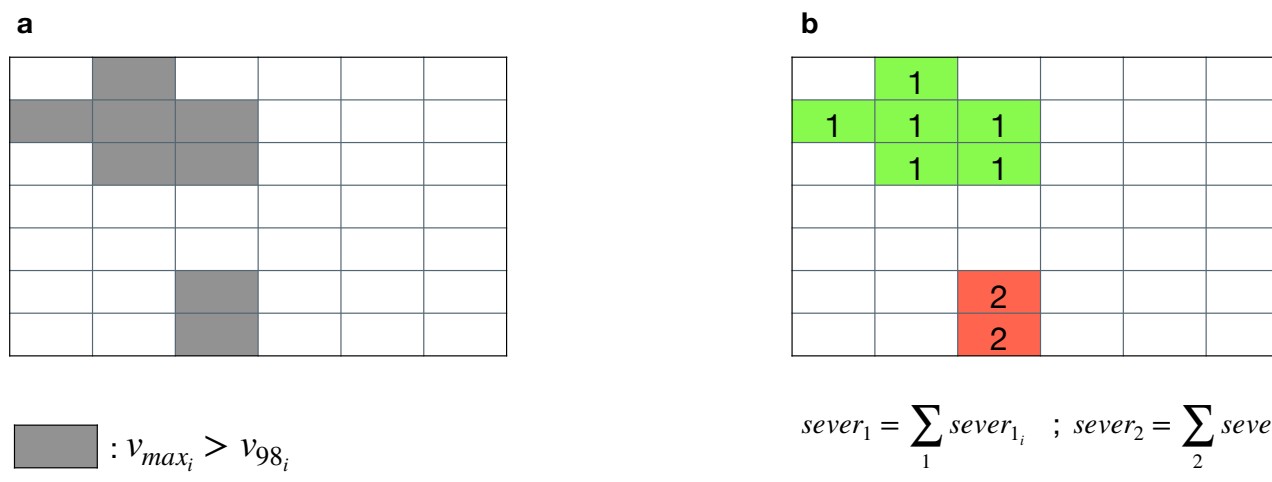

**Figure 2.** Visual depiction of how 10 m wind footprints are identified. (a) To calculate values of severity on a given day, the daily maximum wind speed for each grid cell within the target region is calculated. Then, grid cells where the 10 m daily maximum wind speed has exceeded the local 98th percentile are selected (grey grid cells; there may be no such cells for a given day). (b) Last, connected regions of exceedances are found (green and red regions). Different connected regions are investigated separately. The value of severity for each region is the sum of values of severity of all grids cells within the region. The daily value of severity is equal to the largest single-region value.

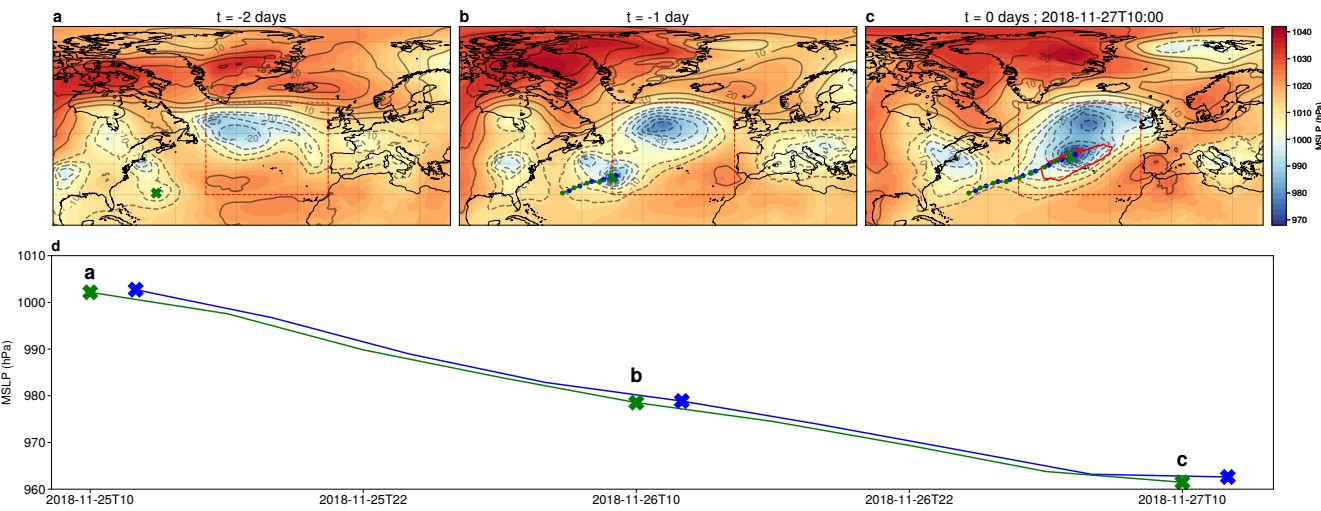

**Figure 3.** MSLP field evolution of the extreme extratropical storm that reached peak 10 m wind speed on 27 November 2018 10 UTC (a–c). The location of the extreme cyclone centre at each time-step is shown as a green cross. The thin dashed red boxes in each panel show the target region. Shading shows absolute MSLP, gray contours show MSLP anomalies relative to the 1950-2020 climatology, starting from $\pm 5$ hPa (dashed for negative anomalies). The wind footprint for the whole day of 27 November 2018 is shown as a thick red contour in (c). Tracks from Pinto et al. (2005) applied on ERA5 (blue dashed line) and manually obtained tracks (green dashed line) are shown in (b,c). MSLP evolution at the cyclone centre from using the two tracking methods is shown in (d), with the green crosses corresponding to the ones in (a–c) and blue crosses showing the closest time available from storm tracks by Pinto et al. (2005).

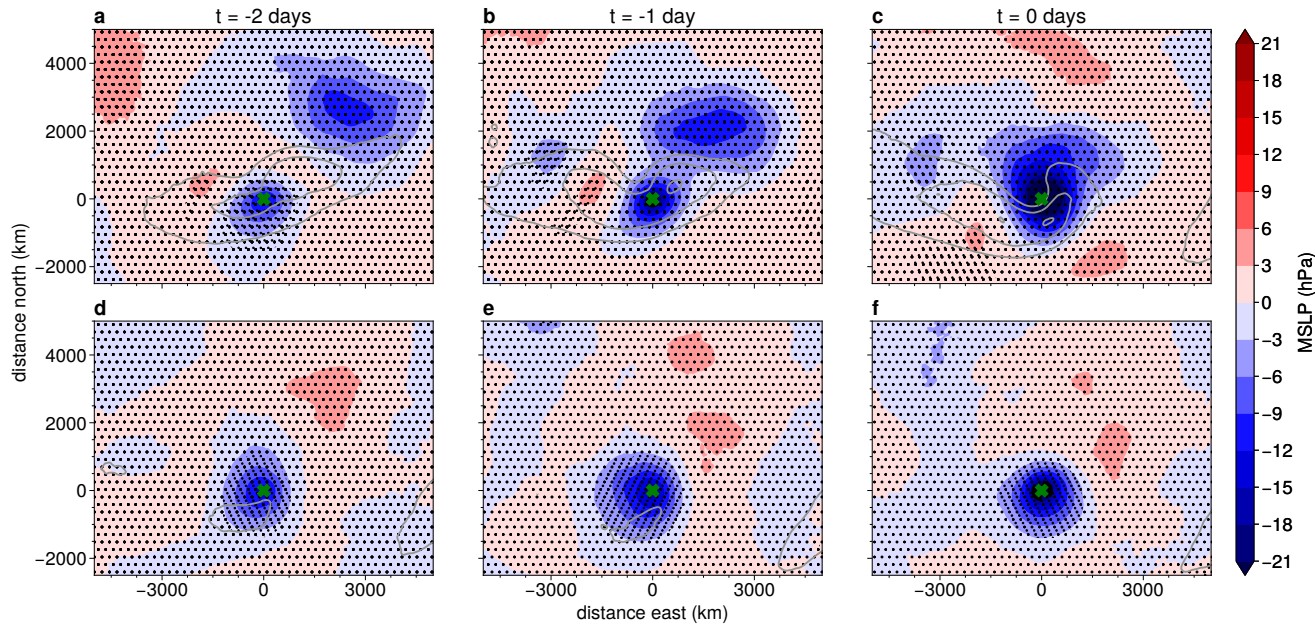

**Figure 4.** Composite MSLP anomalies relative to the 1950-2020 climatology for top extremes (a-c) and moderate extremes (d-f), centered on the cyclone locations from $t = -2$ days to $t = 0$ days. Lags are relative to the time of maximum 10 m wind speeds on the day with maximum severity. Green crosses denote locations of top- and moderate-extreme cyclone centers. Black dots show areas where MSLP anomalies are statistically significant at the 1 % level computed with Monte-Carlo sampling (150 samples, each made from averaging 99 and 117 samples for top and moderate extremes, respectively) and corrected for false discovery (Wilks, 2016). Grey contours show the values of 250 hPa wind speeds, starting from 30 ms$^{-1}$ and increasing in steps of 10 ms$^{-1}$ .

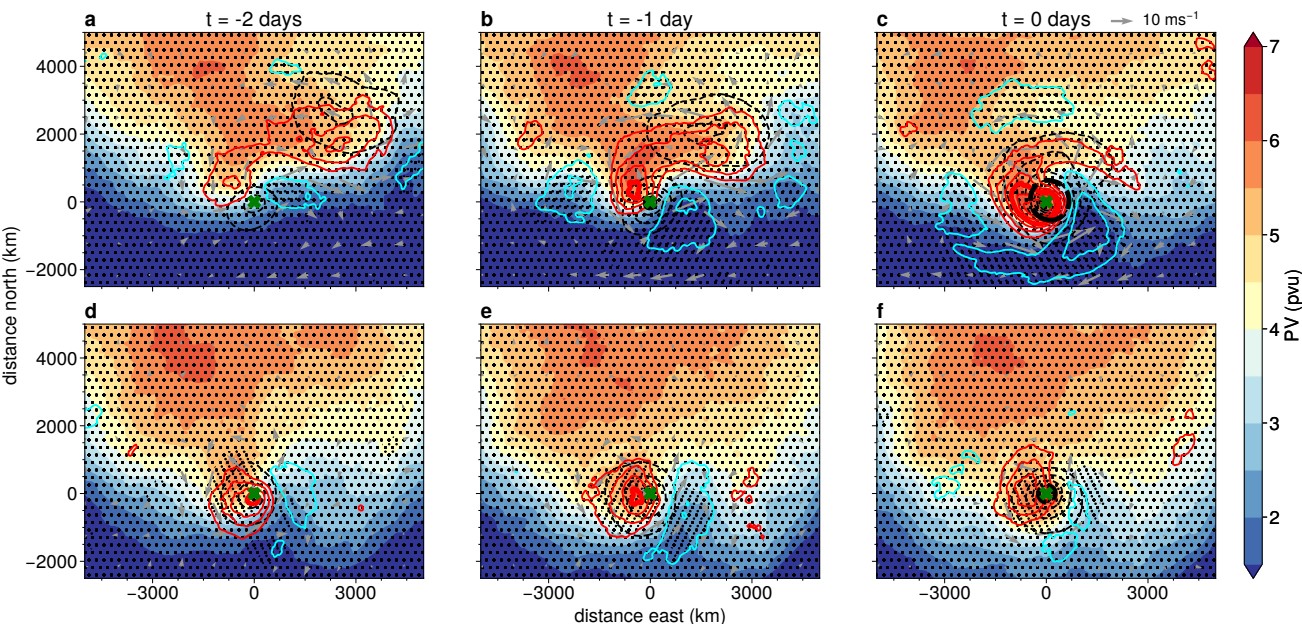

**Figure 5.** Upper-level composites for top extremes (a-c) and moderate extremes (d-f), centered on the cyclone locations from $t = -2$ days to $t = 0$ days. Colors show upper level PV (200-300 hPa mean) values, while red and blue contours show positive and negative upper level PV anomalies relative to the 1950-2020 climatology, respectively, starting from $\pm 0.5$ pvu, with thick contours at $\pm 2$ pvu. Black dots show areas where upper-level PV anomalies are statistically significant at the 1 % level, computed as in Figure 4. Grey arrows show 250 hPa wind speed anomalies from climatology. Black contours MSLP anomalies relative to the 1950-2020 climatology, starting from $\pm 5$ hPa (dashed for negative anomalies) with thick contours at $\pm 20$ hPa . Green crosses have the same meaning as in Figure 4.

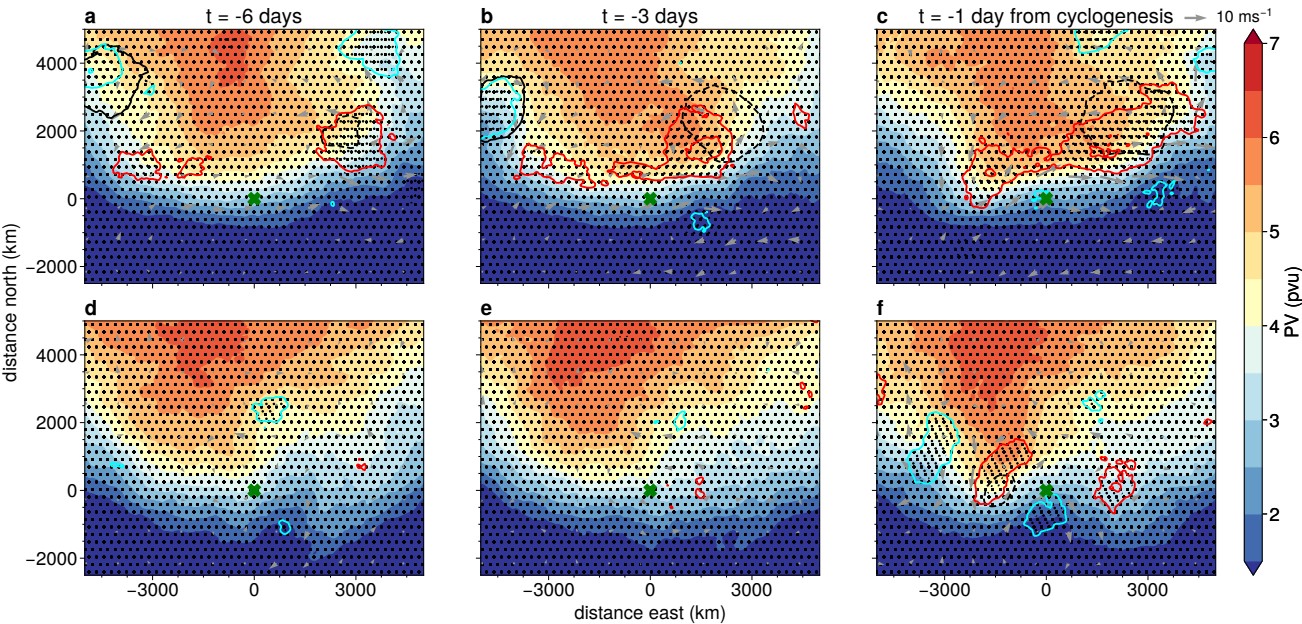

**Figure 6.** Composites centered at the locations of cyclogenesis of top extremes (a-c) and moderate extremes (d-f). Lags are relative to the time of cyclogenesis. Colors, contours, crosses and dots have the same meaning as in Figure 5.

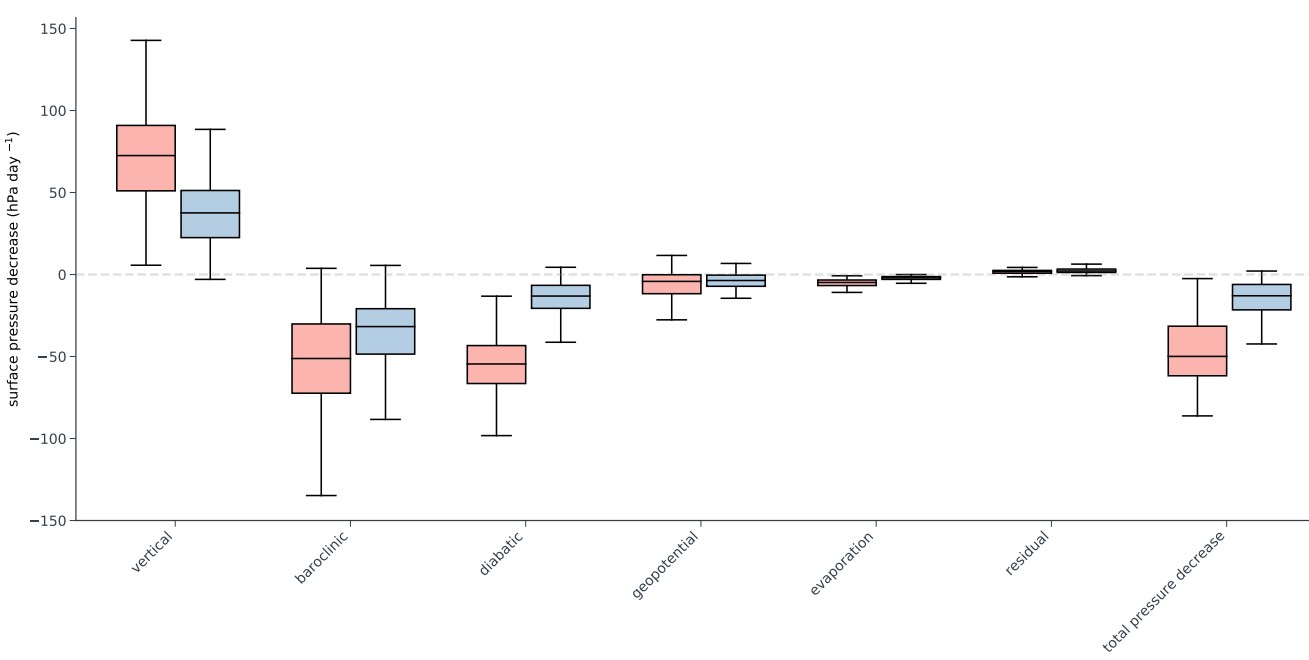

**Figure 7.** Box plots of contributions of each term of the pressure tendency equation to pressure decrease for top extremes (red boxes) and moderate extremes (blue boxes). Boxes show interquartile ranges, black horizontal lines in each box show a median. All terms are averaged over the two days up to the time of maximum 10 m wind speed.

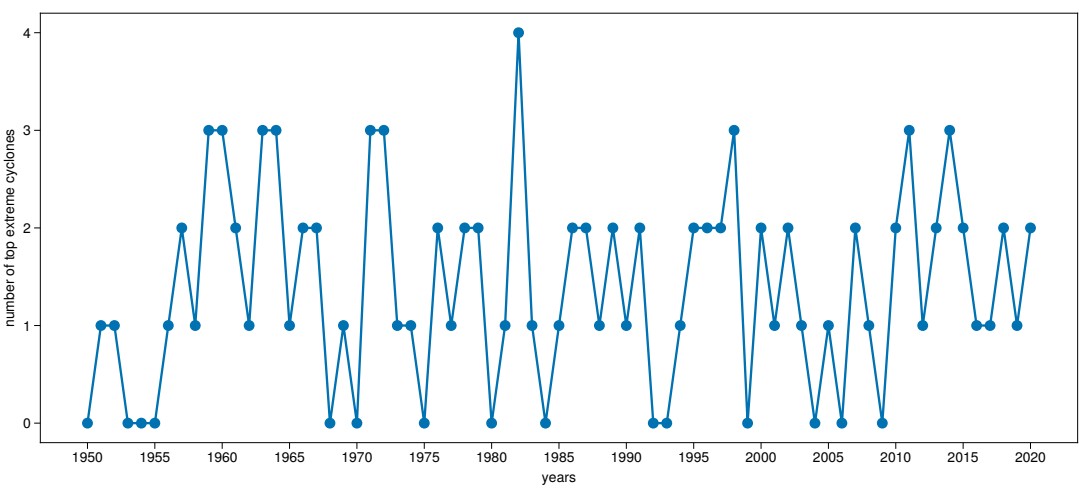

**Figure A1.** Time series of the number of top extreme cyclones in each year from 1950-2020.

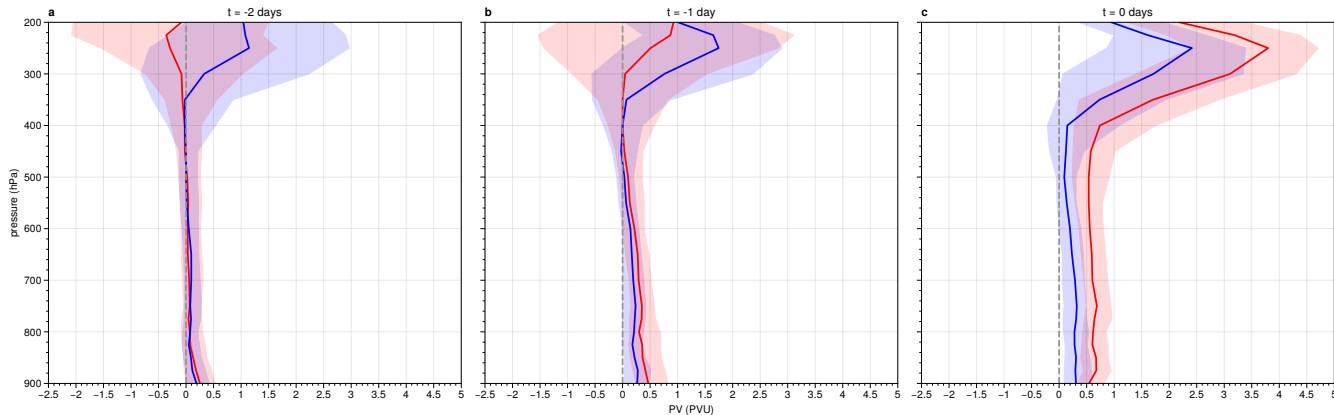

**Figure A2.** Median of PV anomalies from 1950-2020 climatology averaged at all available ERA5 pressure levels from 900 hPa to 200 hPa in a circle with a radius of 300 km around cyclone centers of top extremes (red lines) and moderate extremes (blue lines). Shading shows interquartile ranges.

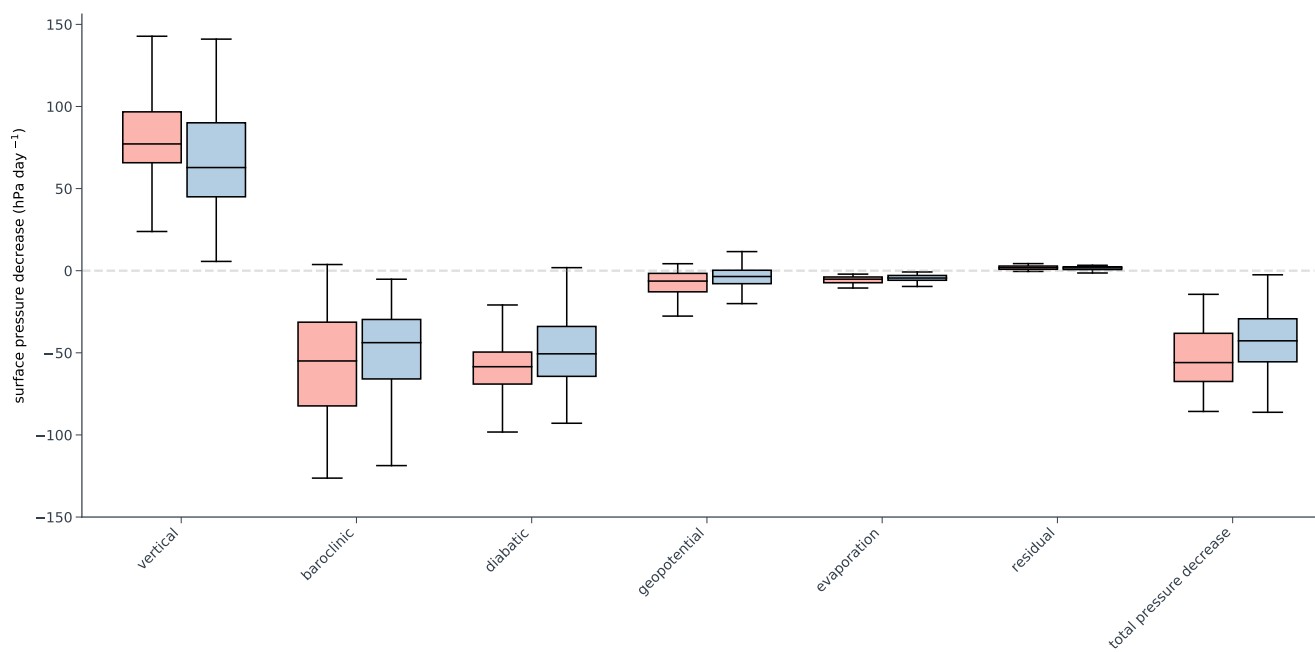

**Figure A3.** Box plots of contributions of each term of the pressure tendency equation to pressure decrease for top extremes that occurred from 1950-1985 (blue boxes) and top extremes that occurred from 1986-2020 (red boxes). Boxes have the same meaning as in 7. Diabatic contribution and total pressure decrease are the only terms that show statistically significant difference between the two groups at the 95 % confidence level according to the Wilcoxon signed rank test.

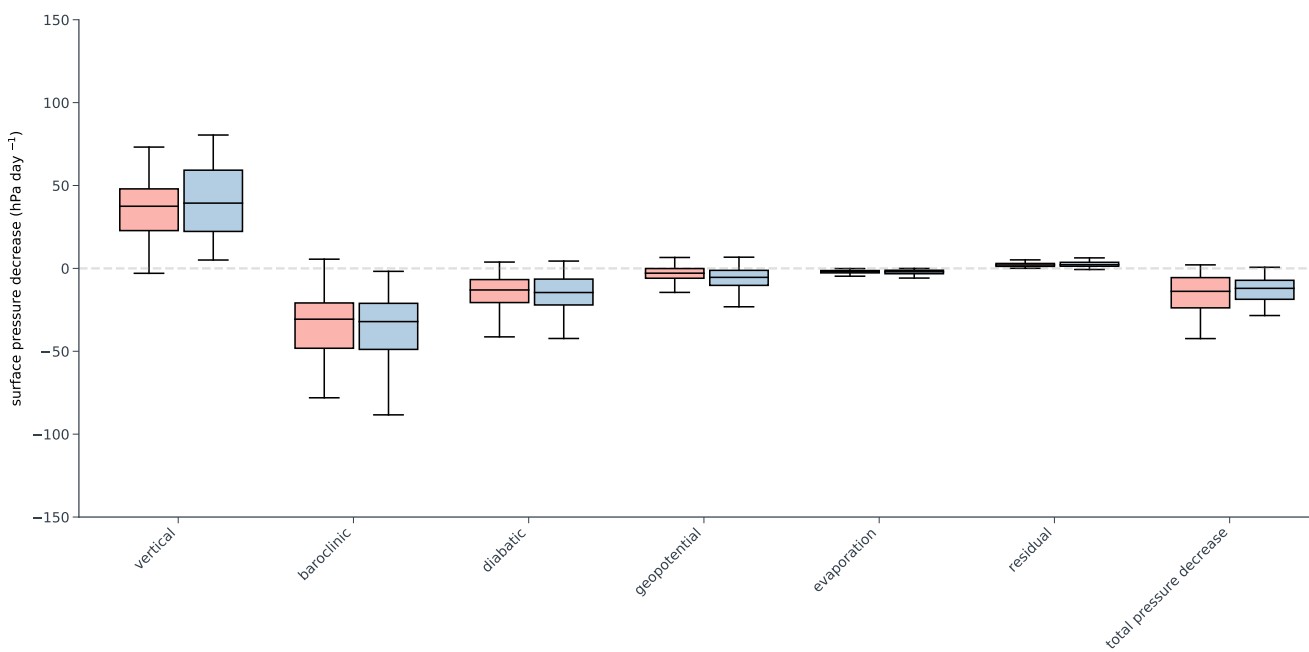

**Figure A4.** Box plots of contributions of each term of the pressure tendency equation to pressure decrease for moderate extremes that occurred from 1950-1985 (blue boxes) and moderate extremes that occurred from 1986-2020 (red boxes). Boxes have the same meaning as in 7. No terms that show statistically significant difference between the two groups at the 95 % confidence level according to the Wilcoxon signed rank test.