# Peer review of "Large-scale perspective on extreme near-surface winds in the central North Atlantic"

_EGUsphere, 2024_

## Author Comment (AC1)

Response to Reviewer 1

We thank the reviewer for the time taken to read through the manuscript and give comments that could help to improve it. Our responses to the comments are found written in blue below.

1.1 The abstract presents some inconsistencies with the rest of the paper. For instance, extratropical cyclones with tropical cyclone origins, which constitute approximately 10% of the top 1% extreme wind events, are excluded from the analysis. However, this exclusion is not mentioned in the abstract. Please ensure that the content presented in the paper is accurately reflected in the abstract whenever possible.

Thank you for pointing out a valid point. The abstract is now changed to clarify that extratropical cyclones with tropical cyclone origins are not analyzed in detail.

Segments in the abstract that deal with this issue will be re-written as: "Cyclones that cause the top 1% most intense wind footprints are identified. After excluding 16 (14%) of cyclones that originated as tropical cyclones, further analysis is done on the remaining 99 extratropical cyclones ('top extremes'). These are compared…"

1.2 The authors should explicitly specify the large-scale atmospheric factors that influence the formation of cyclone extreme wind speeds, which they intend to investigate. While these factors are briefly mentioned in the introduction, they are not explicitly linked to the research questions. Furthermore, certain factors discussed in the paper, such as Rossby wave breaking, are not mentioned in the abstract (as highlighted in the previous point).

Thank you for the comment. We will add the following text to the introduction to link large-scale factors to the goal of our analysis:

"…with them? We investigate how these cyclones differ from weaker cyclones as regards the synoptic-scale features present during their development, their connection with the upper-level potential vorticity fields and their anomalies, as well as the strength of the eddy-driven jet with which they interact. Additionally, we perform surface pressure tendency analysis to quantify the factors behind deepening of top-extreme and moderate-extreme cyclones. "

Additionally, a new sentence will be added to the abstract:

"There is also an indication of cyclonic Rossby wave breaking preceding the top extremes."

1.3 It is important to better emphasize the motivation behind the study. While the authors correctly mention that previous literature focused on extreme wind speeds over land due to

concerns about loss of life and property associated with cyclone extremes, it is necessary to articulate why studying cyclone extreme wind speeds over the ocean is valuable. Aside from practical considerations, such as the absence of topographical features introducing noise in near-surface fields, what is the primary motivation for this aspect of the study? Clarifying this will enhance the paper's context.

We have tried to communicate the motivation behind our study by dedicating a whole paragraph in the Introduction to the reasons why studying extreme winds over the ocean is valuable (paragraph between lines 47-58, the same one referenced in the reviewer's comment). There are several reasons were mentioned in the paragraph, which probably obscured the primary reason for choosing this region for our study. Namely, the reason being that the climatological peak of the storm track is over the ocean and over our region of interest. This provides an opportunity to illuminate how the surface wind extremes form over the peak storm track region. This could then be valuable for comparing these cyclones to land falling ones which typically form at the end of the North Atlantic storm track. While this is a primary motivation, other mentioned reasons (removing the effects of topography and land surface properties, being able to compare driving mechanisms of cyclones that cause extremes over ocean to those causing extremes over land…) are also important and some of them could, even if they are not further investigated in this study, be useful for members of other closely related scientific communities (like those studying marine local ecosystems, as mentioned in the manuscript). To make all of this more straightforward, we have made changes to the passage in the Introduction mentioned:

"Focusing on extreme windstorms over the ocean provides the opportunity to study cyclones that cause extreme 10 m winds in the region of peak of cyclone track frequency. An analysis of this kind is also useful to compare mechanisms driving extreme windstorms over the bulk of the oceanic storm track, and over Europe, which is at the end of the North Atlantic storm track. Moreover, the chosen target region provides a larger sample of intense windstorms than if focussing on land regions, which is an important aspect to consider when studying any extreme event. An additional reason for choosing an ocean region is that it removes the sometimes…"

To further strengthen the reasons why a study like this one is well motivated, we have added an additional, more practical reason that we have not listed before to the same paragraph as the following sentence:

"... On a more practical note, offshore infrastructure and busy shipping routes over the North Atlantic can be severely affected by extreme winds, resulting in sizeable insured losses (Cardone et al, 2015). Finally, …"

1.4 I find the use of the destructiveness index somewhat an unnecessary complication, as it is typically derived for land-based scenarios. I appreciate the motivation behind its use is more leaning towards ranking the extremes rather than evaluating associated potential insurers' losses. Therefore, it is crucial to provide a clearer explanation of the study's purpose, whether it serves scientific, practical, or both purposes. Perhaps consider using the term "severity index" instead to align with the study's objectives.

As the reviewer has pointed out, the main reason behind using this index is to rank the extremes. This index, in the form that we have used, was named as destructiveness index in one of its first uses in Klawa and Ulbrich, 2003. In the following years, many authors have referred to the similar version of the index that we have used as a "meteorological index" and added some modifications to it (mainly the ones that account for population density which does not apply in this case) before forming an index which was then called destructiveness or loss index (for example in Pinto et al. 2012; article suggested by the reviewer in comment 1.10, Leckebusch et al. 2008, is also relevant). Even though some insured losses could occur in the region studied here (as mentioned in the reply for the previous comment), thus making this study having some practical purposes, the main motivation behind using the index in this form was to rank the cyclones which would then be studied and not assessing their potential for causing insured losses.

We agree that calling this index destructiveness index could be misleading and for that reason we agree to refer to it as a "severity index" in a new version of the manuscript. We will rewrite the manuscript so that all the references to destructiveness/destructiveness index are now references to severity/severity index. Figure 2 will also be changed to reflect a change in the name of the index used.

1.5 Another crucial element missing is a discussion of how the choice of cyclone tracking method used may impact the results. There may be significant variations in results, as vorticity-based methods often detect cyclones differently when compared to pressure-based methods (as demonstrated by Neu et al. 2013). Including such a discussion in Section 3 would strengthen the paper.

We agree that different tracking methods could impact the detection of cyclones. As we have tracked the cyclones based on MSLP fields, the alternative approach could have been to use the low-level vorticity. However, we do not think that this approach would have yielded substantially different conclusions and findings in this particular case of tracking the cyclones associated with the most extreme near-surface winds (top extremes). Some of the main original reasons for developing cyclone tracking algorithms based on vorticity fields were to avoid the bias favoring slower and deeper cyclones that pressure-based tracking algorithms have (Sinclair 1994). As the initial detection of cyclones in this manuscript is done in relation to a MSLP minima at the time of the maximum 10-m wind speed, i.e. at the time where top extremes are already deep and developed, bias which favors deeper cyclones should not be relevant. The

other bias of tracking based on MSLP fields, the one which favors slower cyclones, is more of interest since top extremes do tend to move fast as they move along the strong jet. However, the effects of this bias should be greatly reduced when using 1-hourly MSLP fields to perform the tracking compared to 6-hourly MSLP fields that are commonly used. Fast-moving cyclones can cover much greater distances during 6-hourly time-steps than over 1-hourly time-steps, thus making their detection harder if a cyclone moves rapidly. Using 1-hourly fields, however, makes it much easier to "keep track" of the cyclones, as the box within which we search for MSLP minima in the previous time-step covers a distance much larger than that which even the fastest moving cyclones can cover in an hour. Figure 3e in Neu et al. 2013. shows that the fastest moving cyclones identified by all tracking algorithms travel with speed from 60 km/h to 70 km/h which is less than what is needed to exit out of our box of interest within one timestep. An additional difference that could come up as a result of different approaches with tracking is a possible equatorward shift that is noted to occur in some cases when vorticity-based algorithms are used, as noted in Neu et al. 2013. Since our main figures are cyclone-centered composites, even if there was a systematic shift of the center of Figures 4-6, there is no reason to expect that it would be so big to obscure the large-scale patterns show in those Figures. Finally, the differences in locations of deeper, more intense cyclones were found to be smaller in tracking intercomparison studies (Neu et al. 2013). To conclude, although we do agree that different tracking methods can produce different cyclone tracks, we do not think that they would in this particular case make results and conclusions significantly different.

Although we do not think that tracks of top extremes would be significantly affected by using a different tracking algorithm, one part where we believe that these distinctions could matter is in the identification of moderate extremes that are more directly identified from storm tracks. Because we employ a criterion that cyclones in moderate extreme group should have existed for at least 2 days before causing strong winds in the target region and since storm tracking algorithms could differ in finding the time of cyclogenesis for weaker systems, using a different tracking algorithm could influence the number of cyclones constituting the moderate extreme group.

As we find this discussion relevant, we will add a shortened version of it to the manuscript in Section 3, as has been suggested by the reviewer.

1.8 There are some concerns regarding the interpretation of the changes in the budget terms of the pressure tendency equation. Specifically, it is not entirely clear from the presented results whether the diabatic contributions are substantially larger in the top extreme cases relative to the moderate extremes compared to the baroclinic contribution group. To strengthen the robustness of the findings, it is advisable to incorporate non-parametric tests to evaluate the results rigorously.

Note: The comment following the comment 1.5 in the review posted on WCD's website was numbered as 1.8. We have assumed that this has occurred in the process of formatting the

review and decided to stick to the comments numbered as in the review to make the responses easier to follow.

We thank for the comment and agree that a statistical test is necessary to strengthen the robustness of the findings and we have for that reason used the two-sample t-test that showed that for vertical, baroclinic and diabatic terms, the mean values of top extremes are statstically different from moderate extremes (Page 9 line 263). As this is a parametric test, with the assumption that data should follow a normal distribution which might not be a valid assumption for our data, we have also applied non-parametric Wilcoxon Signed Rank test which does not need that assumption. Results after performing this test show that there is a statistically significant difference between the medians of baroclinic, diabatic, vertical and total pressure decrease terms at the 5% level.

Comments about this will be added to the revised version of the manuscript in Section 4.3 and previous parts with t-test will be replaced with those that mention Wilcoxon Signed Rank test.

1.9 An important aspect that appears to be missing is an exploration of how the identified signal varies within the historical period. This is particularly relevant considering the significant changes in our warming climate. It would be valuable to assess how the results compare between the periods 1950-1985 and 1985-2020, especially in terms of the magnitude of the difference between baroclinic and diabatic contributions to the pressure tendency equation. The latter period experiences a human-induced warmer climate signal, and investigating this aspect would greatly enhance the paper's relevance and comprehensiveness.

Thank you for raising is a good point that we have not previously considered. We have performed this analysis and the results from it will be included in the new version of the manuscript. In summary, after differentiating the data between the two periods (1950-1985 and 1986-2020) we have indeed found increased contribution of the diabatic term to the total surface decrease in the period occurring under warmer climate. Although there is an increase in the absolute values of the baroclinic decrease as well from 1986-2020, performing the Wilcoxon Signed Rank test found that the differences in medians are only statistically significant for diabatic and total pressure decrease term. Separation of moderate extremes into two periods did not produce any statistically significant difference, thus showing that climate change signal could only be observed for the group of top extremes. We have added a discussion around these results to Sect. 4.3 and Section 5 when talking about possible directions for future studies. Figures showing these results will be added to Supplementary material.

1.10 Finally, the literature is a bit lacking in some areas concerning storminess in Europe and in the North Atlantic region, motivation for studying extreme wind speeds over the ocean, and dependency of cyclone tracks on chosen methodology. Please add the following publications:

- Earl, N., Dorling, S., Starks, M. and Finch, R. (2017) Subsynoptic-scale features associated with extreme surface gusts in UK extratropical cyclone events. *Geophysical Research Letters*, **44**, 3932–3940.

- Feser, F., Barcikowska, M., Krueger, O., Schenk, F., Weisse, R., & Xia, L. (2015): Storminess over the North Atlantic and Northwestern Europe - A Review. Q. J. R. Meteorol. Soc., 141, 350-382, January 2015 B.

- Gentile, E. & Gray, S. Attribution of observed extreme marine wind speeds and associated hazards to midlatitude cyclone conveyor belt jets near the British Isles. *J. Climatol.* **43**, 2735–2753 (2023).

- Hart, N.C.G., Gray, S.L. and Clark, P.A. (2017) Sting-jet windstorms over the North Atlantic: climatology and contribution to extreme wind risk. *Journal of Climate*, **30**, 5455–5471.

- Hewson, T.D. and Neu, U. (2015) Cyclones, windstorms and the IMILAST project. *Tellus A*, **67**, 27–128.

- C. Leckebusch, D. Renggli, U. Ulbrich, Development and application of an objective storm severity measure for the Northeast Atlantic region, Meteorologische Zeitschrift, Vol. 17, No. 5, 2008

- Manning, C., Kendon, E.J., Fowler, H.J., Roberts, N.M., Berthou, S., Suri, D. and Roberts, M.J., 2022. Extreme windstorms and sting jets in convection-permitting climate simulations over Europe. Climate Dynamics, 58(9-10), pp.2387-2404.

- Messmer, M., I. Simmonds, 2021: Global analysis of cyclone-induced compound precipitation and wind extreme events. Weather and Climate Extremes.

- Ponce de León, S. and Bettencourt, J. (2021) Composite analysis of North Atlantic extra-tropical cyclone waves from satellite altimetry observations. *Advances in Space Research*, **68**, 762–772.

- Ulbrich, U., G. C. Leckebusch, J. Grieger, M. Schuster, M. G. Akperov, N. Y. Bardin, Y. Feng, S. Gulev, M. Inatsu, K. Keay, S. F. Kew, M. L. R. Liberato, P. Lionello, I. I. Mokhov, U. Neu, J. G. Pinto, C. C. Raible, M. Reale, I. Rudeva, I. Simmonds, N. D. Tilinina, I. F. Trigo, S. Ulbrich, X. L. Wang, H. Wernli and the IMILAST team, 2013: Are Greenhouse Gas Signals of Northern Hemisphere winter extra-tropical cyclone activity dependent on the identification and tracking methodology? Meteorologische Zeitschrift, 22, 61-68.

Thank you providing the list of articles missing in the literature. All of them will be added as references in the paper and referenced in the text at various sections.

1.11 Line 21: Could you provide more clarity regarding the specific features you are referring to? It may be beneficial to include a reference like Earl et al. (2017) to support your point.

That sentence was changed and a reference to Earl et al. (2017) was added. However, even though they are very important (like we did briefly mention in the discussion), we did not want to add a lot of descriptions of mesoscale features (like for example sting jets or convective lines) of the windstorms in the introduction, since we did not focus on them in the rest of the article.

1.12 Line 23: The phrase "concentrated on structure" lacks clarity. Please consider rephrasing it for better comprehension.

Rephrasing done.

1.13 Line 26: Could you specify which "close relationship" you are referring to? Providing more specific details will enhance the clarity of your statement.

Done! Vague statement "close relationship" was changed to "positive correlation".

1.14 Line 36-37: The transition between these paragraphs seems abrupt. It would be helpful to create a smoother link between the two paragraphs for better flow and coherence.

The first sentence of the paragraph mentioned was changed in a hope to make a transition between the paragraphs smoother.

1.15 Line 51-52: When stating "few studies," it is important to provide references for credibility. Consider citing relevant works such as Ponce de León and Bettencourt (2021) and Gentile and Gray (2023).

Thank you! References to these studies and short descriptions of their content are added.

1.16 Line 59: Please elaborate on the large-scale factors that you intend to investigate, specifically those associated with the formation of cyclone extreme winds.

Done.

1.17 Line 70: It is worth noting that 10-meter winds are diagnostic rather than prognostic variables, influenced by surface characteristics and the surface layer. A brief discussion on how this might impact the interpretation of your results would be beneficial.

Thank you! End of the first paragraph in Section 2 will have a discussion about this point in the new version of the manuscript.

1.18 Line 84: Consider rephrasing "storm destructiveness index" to storm severity index as it may not be the most appropriate terminology, as discussed in major comment 1.4. Refer to

Leckebush et al. (2008) for guidance. Additionally, better ehighlight why ranking extreme wind speeds by their cube is preferable over 10-meter wind speed, given that the focus is not on evaluating insured losses.

As was said in the response to the major comment, destructiveness index was changed to severity index at all places in the manuscript. There are also small changes in second to last paragraph in Section 3.1 which is the paragraph where we try to say why wind cube should be used even when the focus is not on insured losses.

1.19 Line 113: While the scheme itself is robust, it's important to acknowledge that the choice of tracking scheme may still influence the results. Refer to Ulbrich et al. (2013) and Messmer and Simmonds (2021) for insights into how the tracking scheme chosen could affect your findings.

Discussion about his was added to the end of the Section 3.2.

1.20 Line 136: The exclusion of extreme wind events from extratropical cyclones with tropical origins may lead to the omission of a significant number of events. Please provide additional details, including how large-scale features differ between these two types of extratropical cyclones, apart from the temporal lag between cyclogenesis and extreme events.

Done.

1.21 Line 145-147: The intended message in this section is unclear. Please rephrase for better clarity.

We rephrased this part of Section 3.2.

1.22 Line 152: Could you clarify what you mean by "another part of the analysis"? Please rephrase this section to provide better context.

We believe that line referenced was the line 157, as that is the line that contains the above phrases. We have rewritten that sentence to provide a better context.

1.23 Line 160: Consider incorporating the pressure tendency formula and discussing its relevance in relation to the content in Section 3.4.

Pressure tendency formula was added in the manuscript to help in discussion around PTE analysis.

1.24 Line 200: In addition to numerical values, including percentages would provide a more comprehensive understanding of the data.

Done!

1.25 Line 250 and beyond: It would be beneficial to discuss this paragraph in connection with the equation presented in Section 3.4 once it is included.

As was said in the response for 1.23 PTE was added to the manuscript.

1.26 Line 260: To more robustly compare the shape of distributions between the moderate and top extremes group, consider incorporating a non-parametric test for a more robust analysis.

Wilcoxon Signed Rank test was performed and the results that came from it are discussed.

1.27 Line 280 and beyond: An essential aspect missing from the discussion is an exploration of historical variability within the selected time periods (1950-1985 and 1985-2020). Investigating changes in contributions from the pressure tendency equation terms within these periods is crucial, particularly in light of the non-negligible global warming signal within the analyzed time frame.

That is a very useful comment! We have performed this analysis and discussion around it is included at the end of Section 4.3.

1.28 Line 322-324: It would be valuable to discuss this aspect in relation to Hart et al. (2017) and Manning et al. (2022) to provide a more comprehensive context.

Thanks! We added a sentence and the reference to these articles and methods developed in them in order to provide more context about possible directions for further studies.

1.29 Line 339: Your statement regarding the significant differences between moderate and top extremes relative to diabatic contributions to the pressure tendency equation, is strong. To strengthen your argument, consider conducting statistical tests to support your findings, as mentioned in previous comments

Thank you – done!

References

Cardone, V.J., Callahan, B.T., Chen, H., Cox, A.T., Morrone, M.A. and Swail, V.R., 2015. Global distribution and risk to shipping of very extreme sea states (VESS). *International Journal of Climatology*, *35*(1), pp.69-84.

Klawa, M. and Ulbrich, U., 2003. A model for the estimation of storm losses and the identification of severe winter storms in Germany. *Natural Hazards and Earth System Sciences*, *3*(6), pp.725-732.

Leckebusch, G., Renggli, D. and Ulbrich, U., 2008. Development and application of an objective storm severity measure for the Northeast Atlantic region. *Meteorologische Zeitschrift*, *17*(5), pp.575-587.

Neu, U., Akperov, M.G., Bellenbaum, N., Benestad, R., Blender, R., Caballero, R., Cocozza, A., Dacre, H.F., Feng, Y., Fraedrich, K. and Grieger, J., 2013. IMILAST: A community effort to intercompare extratropical cyclone detection and tracking algorithms. *Bulletin of the American Meteorological Society*, *94*(4), pp.529-547.

Pinto, J.G., Karremann, M.K., Born, K., Della-Marta, P.M. and Klawa, M., 2012. Loss potentials associated with European windstorms under future climate conditions. *Climate Research*, *54*(1), pp.1-20.

Sinclair, M.R., 1994. An objective cyclone climatology for the Southern Hemisphere. *Monthly Weather Review*, *122*(10), pp.2239-2256.

---

## Author Comment (AC2)

Response to Reviewer 2

We thank the reviewer for a thorough reading of our manuscript and for giving the valuable comments that could improve the quality of the manuscript. Our answers to all the points raised on WCD's website are found below written in blue.

1. I wonder how specific you are on differentiating your "footprints". You state that if they are not connected in your analysis region then they are treated as separate. However, what if the situation arises that they are connected outside of the analysis region as part of a larger wind anomaly, how would this work and is this considered?

Thank you for this question! If a situation arises so that wind exceedances over the 98$^{th}$ percentiles are forming a connected region with parts outside of the target geographical region, those parts would not be considered. While this choice could potentially influence the ranking of the storms, we made it to be sure that the focus is on the extreme winds over the ocean in the central Atlantic. Extending the eastern boundaries in these cases would bring in the influence of European continent, extending it to the north would include the region under the stronger influence of Greenland. Extending the boundary to the south would start including more systems of tropical origins, while extending it to the west would come closer to the region of cyclogenesis and potentially move the focus away from the region where the peak of the storm track is. Therefore, we have decided to only consider wind exceedances in the target geographical region. We have, although indirectly, at one point considered the area outside of the target region by looking at wind exceedances over Europe for individual cases of some of the strongest storms from top extreme group. However, wind exceedances over Europe for those storms were substantially smaller compared to historical European windstorm events, which further influenced our choice to restrict our analysis to the target region only.
We added a sentence that clarifies this in the Section 3.1.

2. Throughout you use ERA5 anomalies from the 1979-2020 climatology, yet your features are identified from 1950-2020. Why this choice? Surely your climatology should match the time period which your events are taken from?

We agree with you and you are absolutely right that climatology used should be the one from 1950-2020. The only reason for using the 1979-2020 climatology was practical since at the time when we performed the analysis, ERA5 data from 1950-1978 was not fully processed. Since those datasets are available now, all the figures where we have previously used 1979-2020 climatology (Figures 1, 3-6) will appear as new figures with 1950-2020 climatology used in a new version of the manuscript. Although did not cause any important quantitave or qualitative differences that could significantly change our previous results or discussion.

3. When are your top 1% events found in the event data? Do these align with times of historically high EU windstorm activity (1990s) or are they regularly interspersed throughout the historical record. Some information on this would be of interest to the readers of this manuscript.

Thank you for this question! Figures below show numbers of top 1% of events in each year and in each decade.

[Figure]

[Figure]

As can be seen from the Figures above, top 1% of events seem to be regularly interspersed through the period and there is no apparent trend in their occurrence. One thing that can be noticed is that the year 1999 which had several big windstorms over Europe has zero storms in

top 1% strongest storms in the central Atlantic. There also seems to be a lack of the apparent trend when looking at the total seasonal severity over the whole target region (summed value of severity index for all days and all grid cells in the season, no matter if regions are connected or not). This can be seen at the Figure below.

[Figure]

Short information about the things mentioned above will be presented in the new version of the manuscript in the Section 3.2, especially since this question potentially opens a door for further research on the connections between extreme European and central Atlantic windstorms.

4. L290-291 - you talk about how the PV field is representative of Rossby wave breaking composites, does it not make sense to demonstrate this yourself with composites of the RWB field?

We agree that objective ways of determining whether RWB is associated with these events would be more rigorous (for example the ones used in Wernli and Sprenger, 2007; Barnes and Hartman, 2012; Gomara et al, 2014), but we have chosen not to apply them because of the limited number of cyclones present in our analysis. Objective methods work well when the main interest is analyzing climatology of many wave breaking events and quantifying their characteristics (for example the intensity of RWB). However, since we analyzed a relatively small number of top extremes compared to the broader climatology of objectively identified RWB published in previous studies, our goal was to not to make any quantitative statements. Instead, we relied on qualitatively identifying the ingredient necessary for physical interpretation of RWB (i.e. the overturning of PV gradient) and since we found it to be present, we decided to make a comment about it like was done in some previous similar studies that also did not use objective methods of finding RWB (like Hanley and Caballero, 2012).

Additionally, the focus on a small number of events provides the opportunity to check the PV fields of individual events and subjectively find RWB events. It also gives more confidence that composites of PV field will be representative of RWB.

However, since we did not choose to use any objective method to characterize RWB and because identification of RWB events was not the main objective of the analysis, we decided to use the wording that makes that clear (line 237 in the old version of the manuscript: "… structure reminiscent of cyclonic Rossby wave breaking") and restrained from using strong wording in this case.

5.  You only show upper-level PV in your composites. It would be good to also see composites of lower-level PV. As you state it is likely that your top1% cyclones are somewhat frontal in nature, and so the lower-level PV anomalies may help strengthen that argument and provide further distinction from your bottom 10% category.

This is a good suggestion! Composites of low-level PV (mean between 650-900 hPa, as for example in Čampa and Wernli, 2012) anomalies centered at the locations of top-extreme and moderate-extreme cyclones can be seen below. Since low-level PV field has more noise than upper-level PV field, the area around the cyclone centers shown is smaller than in the other composite figures in the manuscript. As can be seen, the main difference between the two groups lies in the intensity of the lower-level PV anomaly, with top-extreme cyclones having stronger positive low-level PV anomaly associated with them.

[Figure]

To emphasize that the main difference when it comes to lower-level PV anomalies between the two groups of cyclones is a quantitative difference, the figure below shows the median vertical profile of PV anomalies at levels between 900 and 200 hPa at t=0 days, with the shading showing upper and lower quartiles. Anomalies have been calculated in the area with the radius of 300 km around the cyclone center. The figure below emphasizes that a difference in PV anomalies at t=0 days between the groups stretches down to the surface and is not only confined to the upper-level PV anomalies as the current version of the manuscript might suggest. This finding is also in line with the previous findings that cyclones associated with stronger winds have stronger positive PV anomalies at both lower- and upper-levels (Čampa and Wernli, 2012). We plan to include the figure below to a supplementary material with a short discussion around it in the new version of the manuscript.

[Figure]

6. L145-148 – In the bottom 10% category you have 117 events, which is very similar in number to those from the top 1%. How is this the case considering there should be 10x the number of events? Is this because most of them are not associated with ETCs or TCs? Please clarify this in the text

Thank you for this comment. While the number of days in the bottom 10% category is indeed 10x the number of days in top 1%, the events have the similar number because of the extra condition added when identifying moderate extreme events. For them, an additional criterion that cyclones existed for at least 2 days before they caused footprint of wind exceedances in the target region was added. Because of this, there is a decrease in number of moderate events. This decrease is perhaps not a surprise when we consider that around 40% of all cyclones have

a **total** lifetime shorter than 2 days (Neu et al. 2013) and that this condition further requires that cyclones had their cyclogenesis at least 2 days before they came into the region where the bulk of the North Atlantic storm track is. The purpose of this added condition was to make comparison between top-extremes and moderate extremes more meaningful so that we could make the same composites centered at the same time steps (as top extremes typically had their cyclogenesis 2 days prior to the extreme 10-m wind events). This criterion was mentioned in the old version of the manuscript (L145-147), however we agree that it is not clear that this has caused lower number of the events in moderate extreme group. We will, therefore, clarify this in the new version of the manuscript in second to last paragraph in Section 3.2.

7. L136 - are all of your top 1% events associated with ETCs or TCs? Or did you have to discard some that did not

When we were identifying top 1% of the events, all of them could be eventually associated with ECTs or TCs. We were always able to find a MSLP minima connected to the 10 m wind speed maxima and track it back in time. When we found that tracks match those from HURDAT dataset, we have associated them with tropical to extratropical transitions and all the rest of the events were put in the top 1% category. A sentence in third to last paragraph in Section 3.2 was added to make this clear.

- Equation 1: You need a term to state that D is only calculated for gridpoints where v_i>v_98i

    Thank you – a term is added.

- L90 - is the climatology used to create the 98th percentile just Oct-Mar or is this annual?

    It is the extended winter (Oct-March) climatology  from 1950-2020 in this case. The clarification addressing this is added at this place in the manuscript.

- Section 3.2 - i'm a little confused as to what time you used for the tracking (6hour or 1hour). This section could do with some re-writing to clarify this.

    Tracks used for top-extremes are 1-hourly tracks, while those used for moderate extremes are 6-hourly. New version of the manuscript has made this explicit in Section 3.2.

- L111 - proxy to relative vorticity - please change

Thank you – done!

- L161 - you have not yet shown that the surface MSLP decreases in this timeframe

That is a good point, we have changed this sentence to address this.

- L163 - Most cyclones (again you have not yet shown anything for your cyclone analysis set)

Thank you – changed.

- L216-218 - i would move this paragraph up, or at least talk about the jet streaks first before you start discussing the structure of the PV anomalies

We have moved this paragraph up and changed the order in which we talk about jet streaks and the PV anomalies.

References:

Barnes, E.A. and Hartmann, D.L., 2012. Detection of Rossby wave breaking and its response to shifts of the midlatitude jet with climate change. *Journal of Geophysical Research: Atmospheres*, *117*(D9).

Čampa, J. and Wernli, H., 2012. A PV perspective on the vertical structure of mature midlatitude cyclones in the Northern Hemisphere. *Journal of the atmospheric sciences*, *69*(2), pp.725-740.

Gómara, I., Pinto, J.G., Woollings, T., Masato, G., Zurita-Gotor, P. and Rodríguez-Fonseca, B., 2014. Rossby wave-breaking analysis of explosive cyclones in the Euro-Atlantic sector. *Quarterly Journal of the Royal Meteorological Society*, *140*(680), pp.738-753.

Neu, U., Akperov, M.G., Bellenbaum, N., Benestad, R., Blender, R., Caballero, R., Cocozza, A., Dacre, H.F., Feng, Y., Fraedrich, K. and Grieger, J., 2013. IMILAST: A community effort to intercompare extratropical cyclone detection and tracking algorithms. *Bulletin of the American Meteorological Society*, *94*(4), pp.529-547.

Wernli, H. and Sprenger, M., 2007. Identification and ERA-15 climatology of potential vorticity streamers and cutoffs near the extratropical tropopause. *Journal of the atmospheric sciences*, *64*(5), pp.1569-1586.